# Meaning Making as a Lifebuoy in Dementia Caregiving: Predicting Depression from a Generation Perspective Using a Fuzzy-Set Qualitative Comparative Analysis

**DOI:** 10.3390/ijerph192315711

**Published:** 2022-11-25

**Authors:** Vivian Weiqun Lou, Clio Yuen Man Cheng, Doris Sau Fung Yu, Daniel Fu Keung Wong, Daniel W. L. Lai, Alice Ming Lin Chong, Shuangzhou Chen, Kee Lee Chou

**Affiliations:** 1Department of Social Work and Social Administration, The University of Hong Kong, Pokfulam, Hong Kong Island, Hong Kong, China; 2Sau Po Centre on Ageing, The University of Hong Kong, Pokfulam, Hong Kong Island, Hong Kong, China; 3School of Nursing, LKS Faculty of Medicine, The University of Hong Kong, Pokfulam, Hong Kong Island, Hong Kong, China; 4Faculty of Social Sciences, Hong Kong Baptist University, Kowloon Tong, Kowloon, Hong Kong, China; 5Felizberta Lo Padilla Tong School of Social Sciences, Caritas Institute of Higher Education, Tseung Kwan O, New Territories, Hong Kong, China; 6Department of Asian and Policy Studies, The Education University of Hong Kong, Tai Po, New Territories, Hong Kong, China

**Keywords:** meaning making, dementia caregiving, depression, generation, fsQCA

## Abstract

Depressive symptomatology is associated with caregiver burden and poor health outcomes among dementia caregivers. Scholars called for a paradigm shift to focus on positive aspects of caregiving, in particular, meaning making during the caregiving journey. This study draws on the meaning making model and a generation perspective to predict depression among dementia caregivers from two generations, including Baby Boomers who were born between 1946 and 1964 and Generation X who were born between 1965 and 1980, using a configuration approach. Data was collected in a two-wave longitudinal design, from December 2019 to March 2021 in Hong Kong. A fuzzy-set qualitative comparative analysis resulted in six configurations with an overall solution consistency and overall solution coverage of 0.867 and 0.488, respectively. These configurations consist of a different combination of conditions that predict high depressive symptomatology among dementia caregivers in two generations. Specifically, generation is related to five out of six configurations. This study is the first to predict depression among dementia caregivers using a meaning making model from a generation perspective. It advances the understanding of factors contributing to high depressive symptomatology among dementia caregivers from two generations, thus contributing to the future development of generation-responsive assessments, interventions, and policies.

## 1. Introduction

There are 55.2 million people currently living with dementia worldwide [1], and this figure is estimated to triple in 2050 [2]. Although dementia is the seventh leading cause of death globally in 2019 [1], many patients experience a long period of cognitive and functional decline, living with various levels of disabilities for years before dying. Adult children are major caregivers who contributed to dementia care and fulfill the wish of older adults to “age in place” [3], however, it comes with a trade-off in the caregiver’s well-being. Caregiver burden has been found consistently to be associated with a higher risk of depression among caregivers [4,5,6]. This is especially prominent among dementia caregiving as neuropsychiatric symptoms have been observed in 60% to 98% of people with dementia [7,8,9], and are associated with caregiver burden and depression among dementia caregivers [10,11,12,13,14]. Additionally, demographic variables such as gender, age, and education level were associated with depression among dementia caregivers [15,16]. Riding on consistent results in the associations between dementia caregiving and depressive symptoms among caregivers [17,18], further studies to establish their relationships using cross-sectional design are not imperative. On top of that, adult child caregivers are more likely to encounter role conflicts due to job and other caregiving responsibilities [19].

Hong Kong is expected to have more than 330,000 people aged 60 or above living with dementia [20], hence, an increase in the number of adult child dementia caregivers is a logical projection. A recent study conducted in the local context showed that women were most affected by caregiving duties, it is projected that over 60% of caregivers will be female in 2060 [21]. Another study revealed that female working dementia caregivers were more likely to feel stressed from fulfilling dual roles, which impacted mental health negatively [22]. Gender role of caregiving is especially prominent in the Chinese context due to traditional values [23,24].

A meta-review showed that coping strategies involving problem focus, acceptance, and social-emotional support were beneficial to dementia caregivers [25]. Resilience, the capacity for successful adaptation when confronting stressful life events among caregivers was found to be independent of the clinical symptoms of dementia [26], and negatively associated with their depressive symptomatology [26,27]. Apart from seeking support from an external source, scholars have shifted their focus on how caregivers find meaning in their caregiving journey [28,29,30].

### Meaning Making Model

Meaning is defined as the “mental representation of possible relationships among things, events, and relationships” [31]. A phenomenal integrative review of meaning making provided a systematic way to understand how individuals make sense of stressful life events [32]. This is the ability of an individual to transform meaning. According to the meaning making model, individuals possess global meaning to interpret their experiences and consists of beliefs, goals, and subjective feelings. Potentially stressful events, such as dementia caregiving, will challenge the global meaning of individuals. Under this circumstance, individuals will appraise difficult situations and assign meanings to them [33]. This means a specific evaluation of an event. To minimize the discrepancies between situational meaning and global meaning, individuals engage in meaning-making processes, with the results or changes being termed “meaning made” [32]. This is a positive outcome to cope with stressful events and enhance wellbeing. In this sense, when there is a discrepancy between global and situation meaning, the meaning-making processes occur to provide a new appraisal of the stressful events [32]. For this paper, a generational perspective is the proxy of global meaning, dementia caregiving is the proxy of situational meaning, and adaptation to the caregiving role is the proxy of meaning made. Figure 1 shows the conceptual framework of the meaning-making model from a generation perspective.

Meaning making has been found to moderate the associations between patient depression and caregiver burden [34]. Although from an existential point of view, meaning making provides an alternative paradigm for understanding the dementia caregiving experience [35], existing literature neglect the fact that meanings are highly versatile and are influenced by social norms [33]. In particular, global meaning is assumed to be constructed early in life and modified subsequently based on personal experiences [36,37]. Given this in mind, it becomes logical to ask these questions: How do adult child dementia caregivers from distinctive generations differ in making sense of their caregiving experiences, and how does meaning making moderate their depressive symptoms? Are meaning made among caregivers sufficient to counteract depressive symptoms along the caregiving journey?

Baby Boomers were born following World War II and are considered to be the first generation to enjoy a comparatively higher educational attainment and wealth. Family structures and experiences among the Baby Boomer generation evolved in an unprecedented manner, including an increased divorce rate and a soaring number of women entering the workforce [38]. Unlike their counterparts in the West, the Baby Boomers generation grew up in a Chinese cultural context and still treasures the traditional virtue of filial piety and is thus willing to provide caregiving to their parents with dementia [39,40,41]. Generation X (Gen X), composed of the next generation of Baby Boomers, was signified by flexibility, independence, and self-reliant traits. As a generation, Gen X has a different global view as compared to the Baby Boomer generation and places high value on work-life balance which allows them to focus on family as well as personal well-being [42]. Due to the soaring number of dementia caregivers in Hong Kong, it is expected these two generations will be particularly prone to depression due to caregiving duties as well as experiencing aging themselves. In this sense, we purposefully selected Baby Boomers and Gen X in our study [21,22].

There are three major gaps in extant literature. First, there is a lack of a generational approach to understanding dementia caregivers, despite it having been investigated and incorporated into different disciplines, such as marketing [43], technology [44], and healthcare [45]. Second, positive aspects of caregiving such as searching for meaning are insufficient. Recently, several scholars sought to examine differences in caregiver mental health outcomes across generations [46], however, they only focus only on the negative aspects of dementia caregiving. Third, heterogeneity among dementia caregivers was disregarded using conventional analysis strategies. What is lacking and urgently needed is empirical evidence to unveil conditions contributing to depressive symptoms among dementia caregivers.

In this study, we are applying a generation perspective to examine the factors leading to depressive symptomatology among Baby Boomers and Gen X dementia caregivers, traditional analysis strategies are eloquent of an assumption of homogeneity of individuals from the same generation. To unveil different possibilities that contributed to depressive symptoms during the dementia caregiving journey, we should embrace the heterogeneity of dementia caregivers who had different individual experiences from a life course perspective by employing the fuzzy-set qualitative comparative analysis (fsQCA). fsQCA originated in the social sciences field to combine case-oriented and variable-oriented quantitative analysis, it started with the creation of qualitative comparative analysis, drawing on the fuzzy-set theory [47]. Guided by the meaning making model, this study examined the relationships between meaning making, generation, and high depressive symptomatology among dementia caregivers, using the fsQCA [48,49].

## 2. Materials and Methods

### 2.1. Study Design

This was a two-wave longitudinal study of dementia caregivers from two generations, including Baby Boomers who were born between 1946 and 1964 and Gen X who were born between 1965 and 1980. A longitudinal study allows the researcher to be able to detect changes in depressive symptoms of dementia caregivers and explore the underlying mechanisms leading to the changes. In this study, data were collected from December 2019 to March 2021 in Hong Kong. A standardized questionnaire was designed to gather data on dementia caregivers and their care recipients at two time points, with a 12-month interval. Ethics approval was obtained from the Human Research Ethics Committee, The University of Hong Kong as well as the Institutional Review Board of The University of Hong Kong/Hospital Authority Hong Kong West Cluster, respectively. All the participants provided written informed consent for study participation.

### 2.2. Participants

Dementia caregivers who were Hong Kong residents born between 1946 and 1980, assisted at least one of their parents (aged 65 or above and diagnosed with dementia) with activities of daily living (ADL) [50] or instrumental activities of daily living (IADL) [51] for not less than eight hours per week three months prior to the survey [52], able to speak Cantonese, and willing to participate, were included in this study. Those who were challenged by stressful life events in his/her own life (e.g., diagnosed with a life-threatening disease, unemployed, or experiencing a loss of close family members or friends at the time of the recruitment), were excluded [53]. This was assessed on enrolment through a self-reported approach. Participants were recruited through community service units, caregiver associations, and social networks of the research team.

### 2.3. Measures

Outcome measures were collected with each participant at baseline and follow-up by a trained research assistant. The analysis followed the principle of causal asymmetry underlying the configuration theory [48,49]. In this paper, the analysis only included participants who were taking care of their mother or father with dementia, self-reported adaptation to the caregiving role, and completed the questionnaire at both time points. These participants are not affected by dementia themselves but taking care of parents with dementia. All the questionnaires were conducted using the local language, that is Traditional Chinese (in written) and Cantonese (in verbal).

#### 2.3.1. Depressive Symptoms

The main outcome measure for dementia caregivers was depressive symptoms among dementia caregivers using the Patient Health Questionnaire-9 (PHQ-9) [54]. The PHQ-9 is a self-administered 9-item validated measurement, each item is rated on a 4-point scale, ranging from 0 (not at all) to 3 (nearly every day). The potential total score ranged from 0 to 27, scores of 5, 10, 15, and 20 indicated mild, moderate, moderately severe, and severe depression [54]. The PHQ-9 has been validated in the Hong Kong population, with an internal consistency of 0.82 [55]. The PHQ-9 measured at baseline was treated as an independent variable, while the PHQ-9 measured at follow-up was the dependent variable.

#### 2.3.2. Generation

Generation is the proxy of global meaning. It was measured using the year of birth of the dementia caregivers. Those who were born between 1946 and 1964 were categorized as Baby Boomers [56], while those who were born between 1965 and 1980 were categorized as Gen X [57].

#### 2.3.3. Situational Meaning

Situational meaning is the specific evaluation of a potentially stressful event. It was measured by two subscales, loss/powerlessness and provisional meaning, of the Finding Meaning Through Caregiving Scale (FMTCS) [58]. The 19-item sense of losses/powerlessness subscale was composed of three themes that reflect caregivers’ feelings of loss for their family members, feeling of loss about themselves, and feelings of powerlessness attributed to caregiving [35]. Each item was rated on a 5-point Likert scale with a potential total score ranging from 19 to 95. A higher score indicates a higher sense of loss/powerlessness. The 19-item provisional meaning subscale was composed of three themes that assess caregivers’ values on positive aspects of life and caregiving, personal choice, and means that aid caregivers to find pleasures [58]. Each item was rated on a 5-point Likert scale with a potential total score ranging from 19 to 95. A higher score indicates a higher provisional meaning.

#### 2.3.4. Meaning Made

According to the meaning making model [32], meaning made includes an acceptance of a situation and a changed identity. In this study, the situation is dementia caregiving while a changed identity is the acknowledgment of a dementia caregiver role. For this purpose, meaning made was measured using a proxy, adaptation period, with a single item: “Have you adapted to the caregiver role?” For caregivers who have adapted to the caregiving role, they were asked the number of months that they used for adaptation. For others, they were asked to provide reasons for non-adaptation [59]. In this analysis, we only selected caregivers who have adapted to their caregiving roles.

#### 2.3.5. Caregiver Burden

Caregiver burden was measured using the 4-item version Zarit Burden Interview (ZBI-4) [60]. Each item was rated on a 5-point Likert scale ranging from 1 (never) to 5 (nearly always), with a potential total score ranging from 4 to 20. A higher score indicates a higher caregiver burden. The ZBI-4 is extracted from the Cantonese short version of the ZBI (CZBI-Short) which has 12 items. The CZBI-Short has been validated among Hong Kong dementia caregivers, with an internal consistency of 0.84 [61].

#### 2.3.6. Neuropsychiatric Symptoms

Neuropsychiatric symptoms (NPS) were measured using the 12-item Neuropsychiatric Inventory Questionnaire (NPI-Q) [62]. The NPI-Q is a caregiver-based questionnaire in which the caregiver indicated the presence or absence of NPS among their care recipients in the past month. Additionally, the NPI-Q considers the severity of NPS and caregiver distress. Scores on the severity scale ranged from 1 (mild) to 3 (severe) and the scores of the caregiver distress ranged from 0 (no distress) to 5 (extreme distress). The NPI-Q has been validated among caregivers in Hong Kong with an internal consistency of 0.76 [63].

#### 2.3.7. Resilience

Resilience was measured by a 10-item Connor-Davidson Resilience Scale (CD-RISC 10) [64]. Each item was rated on a 5-point Likert scale ranging from 1 (never) to 5 (nearly always), with a potential total score ranging from 5 to 50. A higher score indicates higher resilience. The CD-RISC has been validated among the general population in Hong Kong, with an internal consistency of 0.97 [65].

#### 2.3.8. Demographic Information

Demographic information on dementia caregivers and their care recipients was collected. For dementia caregivers, age, gender, marital status, education level, housing, affordability of caregiving, and working status were included. For their care recipients, age, gender, living condition (i.e., living with a caregiver), the total number of people living together, year of diagnosis of dementia, ADL, and IADL were included, and data were reported by the caregivers. ADL was measured by the Chinese version of the Modified Barthel Index (CMBI) to assess the care recipients’ ability to perform ten personal activities, with a potential total score ranging from 0 to 100 [50,66]. A higher score indicates a higher ability in performing ADL. The CMBI has been developed and validated in Hong Kong, with an internal consistency of 0.93 [66]. IADL was measured by the 8-item Hong Kong Chinese version of the Lawton Instrumental Activities of Daily Living Scale (CIADL), with a potential total score ranging from 0 to 8 [51,67]. A higher score indicates a higher functionality in IADL. Similar to CMBI, CIADL has been locally validated with an internal consistency of 0.86 [67].

### 2.4. Statistical Analysis

Data were summarized using descriptive statistics. Continuous variables were presented using mean with standard deviation, while categorical variables were described using frequencies with percentages. Demographic information and variables of interest among dementia caregivers and their care recipients from two generations, Baby Boomers and Gen X, were compared using independent *t*-tests and chi-square tests for continuous and categorical variables, respectively, with statistical significance set at *p* < 0.05.

#### 2.4.1. Bivariate Analysis of Factors Associated with Depressive Symptoms

The relationships between PHQ-9 at follow-up and selected variables, including generation, sense of loss/powerlessness, provisional meaning, caregiver burden, depressive symptoms at baseline, NPS disturbance, resilience, adaptation period, and NPS severity, were determined using Pearson correlation. A correlation coefficient of 0.10 to 0.29, 0.30 to 0.49, and 0.50 to 1.0 denote small, medium, and large effect sizes [68].

#### 2.4.2. Fuzzy-Set Qualitative Comparative Analysis of Depressive Symptoms

While the development of fsQCA is still in its infancy, the number of researchers across disciplines who use fsQCA has steadily increased [69]. This paper is the first to apply fsQCA to understand the factors contributing to depressive symptomatology among dementia caregivers. One of the main characteristics of fsQCA is the degrees of freedom it provided to allow researchers to examine the causal complexities involved in conjunctural causation [70]. Fundamentally, the aim of using fsQCA is to examine the cases configurationally, meaning that separate parts of a whole picture have to be understood in a case-oriented manner, embedded in a specific context such as in a dementia caregiving journey. In addition, by following the meaning making model, we look into various aspects and combinations of conditions to obtain a comprehensive understanding of the homogeneity within the context. In general, by applying fsQCA and following the meaning making model, this analysis strategy allows us to depict the combinations (also called solutions in fsQCA) to examine the factors leading to depressive symptomatology among dementia caregivers from two generations, while not losing sight of both heterogeneity and homogeneity.

Configurational analysis was performed using SPSS version 27 (IBM^®^, Armonk, NY, USA), QCA package in R version 4.2.0 [71], and fsQCA software version 3.0 at different steps. Key analytic fsQCA steps include analyzing contrarian cases, choosing thresholds, calibrating data, constructing a truth table, sorting the truth table, computing solutions, and identifying predictive validity [72,73].

First, we conducted a contrarian case analysis in SPSS to identify the main effects and contrarian cases (Appendix A). Next, we performed data calibration in R to transform the antecedent conditions and outcomes into a membership score ranging from 0 to 1. Specifically, membership scores of 0 and 1 indicate non-full and full membership, respectively. A score of 0.5 denotes intermediate set membership. In this study, we used the totally fuzzy and relative (TFR) method [74]. Data calibration using the TRF method was based on rank orders, it is particularly applicable for this study as we categorize dementia caregivers into two generations and comprised both ordinal and interval levels data. The TFR technique relied on an empirical cumulative distribution function on the data, and it is best suited to interval-level data. A normalized version by applying a simple transformation to create a membership score was applied to categorical data. We followed prior literature in applying the below formula for data calibration [75]. In which, *E*() is the empirical cumulative distribution function of the data.
$$TFR=\max\left(o,\;\frac{E\left(x\right)-E\left(1\right)}{1-E\left(1\right)}\right)$$

After data calibration, we constructed a truth table with 2*^k^* rows using the fsQCA software, where *k* denotes the number of antecedent conditions. In this study, the truth table involves 2^9^ logically possible combinations of antecedent conditions. The truth table was then minimized by the number of cases smaller than 1, a raw consistency value below 0.75, and a proportional reduction in inconsistency below 0.70 [72], following the Quine-McCluskey algorithm (i.e., minimizing Boolean functions) [76,77,78].

The fsQCA provides three types of solutions, namely complex, parsimonious, and intermediate [48]. The complex solution presents all possible combinations when a logical operation (i.e., choosing the presence or absence/negation of a variable) is applied. The parsimonious solution is a simplified version of the complex solution, it only presents the “core conditions” (i.e., conditions indicate a strong causal relationship with the outcome). The intermediate solution is retrieved following a counterfactual analysis of the complex and the parsimonious solutions, it only presents theoretically plausible counterfactuals [48]. It is important to note that while “core conditions” are presented in both parsimonious and intermediate solutions, conditions that are eliminated in the parsimonious solution but appear only in the intermediate solution are called “peripheral conditions” (i.e., conditions that indicate a weaker causal relationship with the outcome) [48]. In this sense, we identified the “core conditions” by examining the parsimonious solution and included the “peripheral conditions” by referring to the intermediate solution. This strategy of combining the parsimonious and intermediate solutions offers an aggregated view of the findings and provides richer information for interpretation [48].

## 3. Results

### 3.1. Characteristics of Caregivers from Two Generations

Of the 396 dementia caregivers enrolled in this study, a total of 167 were adult child caregivers (42.2%), with 116 (69.5%) of them adapted to the caregiving role. Among these caregivers, 101 were Baby Boomers and 66 were Gen X. The demographic characteristics of these 167 caregivers from two generations were shown in Table 1. The mean age of Baby Boomers and Gen X dementia caregivers were significantly different (*p* = 0.001) as expected. Most of the dementia caregivers were female (79.0%), around half of them were married (50.9%), less than half of them obtained a higher diploma or above (44.3%), and around two-thirds of them were residing in public housing (59.9%), and most of them reposted that they were able to afford caregiving (89.2%). Working status among dementia caregivers from two generations was significantly different, with more Gen X dementia caregivers working at the time of baseline data collection (68.2% vs. 43.6%, *p* = 0.002).

### 3.2. Characteristics of Care Recipients from Two Generations

The demographic characteristics of the 167 care recipients were shown in Table 1. The mean age of care recipients was 84.76 years. Most of the care recipients were mothers (78.4%), and over half of them were living with a caregiver (55.1%). These care recipients had an average of 2.34 (SD = 1.28) people living in the same household, with an average of 5 (SD = 3.25) years of diagnosis of dementia. Regarding daily activities functioning, care recipients had an average of 4.11 (SD = 1.99) score in ADL and 1.60 (SD = 2.11) in IADL. Significant difference in IADL was found between care recipients taking care by two generations of dementia caregivers, IADL performance was better among care recipients of the Gen X dementia caregivers (M = 2.00, SD = 2.46 vs. M = 1.34, SD = 1.81, *p* = 0.047).

### 3.3. Differences in Outcomes among Caregivers from Two Generations

Significant differences between dementia caregivers from the two generations were found in sense of loss/powerlessness, caregiver burden, and resilience. Details have been listed in Table 2. Gen X dementia caregivers reported a significantly higher sense of loss/powerlessness as compared to Baby Boomer dementia caregivers (t(165) = 2.641, *p* = 0.009). In addition, Gen X dementia caregivers reported a significantly higher caregiver burden as compared to Baby Boomer dementia caregivers (t(165) = 2.766, *p* = 0.006). Relatedly, Gen X dementia caregivers reported a significantly lower resilience as compared to Baby Boomer dementia caregivers (t(165) = −2.070, *p* = 0.040).

### 3.4. Bivariate Analysis of Factors Associated with Depressive Symptoms

Results of bivariate analysis of factors associated with depressive symptoms among dementia caregivers were presented in Appendix A. Depressive symptoms at follow-up were positively associated with a sense of loss/powerlessness (r = 0.493, *p* < 0.001), caregiver burden (r = 0.422, *p* < 0.001), depressive symptoms at baseline (r = 0.516, *p* < 0.001), caregiver distress (r = 0.397, *p* < 0.001), adaptation period (r = 0.284, *p* < 0.001), and NPS severity (r = 0.328 *p* < 0.001). Additionally, depressive symptoms at follow-up were negatively associated with resilience (r = −0.238 *p* = 0.002).

### 3.5. Fuzzy-Set Qualitative Comparative Analysis of Depressive Symptoms

The intermedia solutions containing both the “core conditions” and “peripheral conditions” obtained from fsQCA were shown in Table 3. A total of six configurations were obtained, with an overall solution consistency of 0.867, indicating that the configurational combinations for high depressive symptomatology were useful and can serve theory advancement [79]. Furthermore, the overall solution coverage was 0.488, meaning that 48.8% of depressive symptoms among dementia caregivers may be explained by the six configurations. In general, solution 1 was unrelated to generation. Solutions 2–5 were specifically useful in explaining high depressive symptoms among Baby Boomers dementia caregivers, while solution 6 was adopted to explain high depressive symptoms among Gen X dementia caregivers.

**Solution 1:** The presence of—a high sense of loss/powerlessness, high caregiver burden, high depressive symptoms at baseline, high caregiver distress, longer adaptation period, high NPS severity, with the absence of—high resilience, lead to high depressive symptoms at follow-up, regardless of generation and sense of provisional meaning. **Solution 2:** Among Baby Boomers’ dementia caregivers, the presence of—a high sense of loss/powerlessness, high caregiver burden, high depressive symptoms at baseline, longer adaptation period, with the absence of—high caregiver distress, high resilience, and high NPS severity, lead to high depressive symptoms at follow-up, regardless of the sense of provisional meaning. **Solution 3:** Among Baby Boomers’ dementia caregivers, the presence—of a high sense of loss/powerlessness, high caregiver burden, high depressive symptoms at baseline, and high NPS severity, with the absence of—a high sense of provisional meaning, high resilience, and longer adaptation period, lead to high depressive symptoms at follow-up, regardless of caregiver distress. **Solution 4:** Among Baby Boomers’ dementia caregivers, the presence of—a high sense of loss/powerlessness, high caregiver burden, high depressive symptoms at baseline, high caregiver distress, longer adaptation period, and high NPS severity, lead to high depressive symptoms at follow-up, regardless of resilience. **Solution 5:** Among Baby Boomers’ dementia caregivers, the presence of—a high sense of provisional meaning, high caregiver burden, high depressive symptoms at baseline, longer adaptation period, and high NPS severity, with the absence of—a high sense of loss/powerlessness and high caregiver distress, lead to high depressive symptoms at follow-up, regardless of resilience. **Solution 6:** Among Gen X dementia caregivers, the presence of—a high sense of loss/powerlessness, high sense of provisional meaning, high caregiver burden, high depressive symptoms at baseline, high caregiver distress, and high NPS severity, with the absence of—longer adaptation period, lead to high depressive symptoms at follow-up, regardless of resilience.

## 4. Discussion

This study is among the first to use a fuzzy-set qualitative comparative analysis to explore the predictors of depressive symptomatology among adult child dementia caregivers from two generations. It is unique in several aspects. First, this study expands knowledge on adult child dementia caregivers from a generational perspective. Second, it forms the basis of a database of dementia caregivers by collecting and analyzing data using a two-wave longitudinal design. Third, it advances the literature by following the configuration theory and provides a total of six solutions with different combinations of factors that lead to high depressive symptomatology among adult child dementia caregivers. With one unrelated to the generation effect, four targeted Baby Boomers dementia caregivers, while the remaining one focused on Gen X dementia caregivers. The findings provide a deeper understanding of the routes to depressive symptomatology among adult child dementia caregivers from two generations and offer insights into future interventions for adult child dementia caregivers using a generation-responsive strategy.

Our findings indicated that Gen X dementia caregivers reported a higher sense of loss/powerlessness, a higher caregiver burden, and lower resilience than Baby Boomers dementia caregivers at baseline. Although the care recipients of Gen X dementia caregivers were significantly younger as compared to those of Baby Boomers, we should acknowledge the fact that the deterioration rate of the cognitive and functional status of people with dementia does not necessarily depend on their age. Additionally, nearly 70% of Gen X dementia caregivers were working. Taking care of their parents with dementia means they need to struggle between work and caregiving responsibilities, which brings negative impacts on them. For instance, they may need to sacrifice their career to provide care (e.g., switching to a part-time job or forgoing promotion) but at the same time, they would be worried about their financial stability [21,22,80]. Juggling between dual roles will contribute to an increase in the sense of loss/powerlessness, a higher caregiver burden, and a shrink in resilience among Gen X dementia caregivers [81].

This study found that high levels of caregiver burden and depressive symptoms at baseline were core conditions in all six solutions leading to high depressive symptomatology, which is consistent with prior studies [82,83,84]. This result indicates that regardless of generational effect, it is important for future interventions targeting dementia caregivers to emphasize the two major elements: reducing caregiver burden and depressive symptoms. It is interesting to note that although resilience was not presented as a core condition in any of the six solutions, it is important to learn that the absence of high resilience in Solutions 1-3 contributed to high depressive symptomatology among dementia caregivers. Therefore, resilience-enhancing interventions such as mindfulness training [85], meditation [86], and family resilience reinforcement programs [87], may shield dementia caregivers from developing depression, especially among Baby Boomers [88].

Among five solutions (except Solution 2), the presence of high NPS severity in combination with other conditions contributed to high depressive symptomatology among dementia caregivers. This result is consistent with prior literature that showed a relationship between caregiver burden, depressive symptoms, and dementia patients’ NPS severity [83]. The results confirmed the positive association between NPS severity and caregiver distress but also revealed distinctive results from a generation perspective. In particular, high NPS severity may lead to high depressive symptomatology among Baby Boomers’ dementia caregivers in the absence of caregiver distress.

It is interesting to note that meaning-making forms part of the solutions leading to high depressive symptomatology among dementia caregivers, which is consistent with a prior integrative review [29]. Specifically, a high sense of loss/powerlessness contributed to high depressive symptomatology, regardless of the generation of dementia caregivers. It is therefore important to develop interventions that guide and encourage dementia caregivers to search for meanings after adapting their caregiving roles. For example, the Happy Times Card (https://ageing.hku.hk/en/happytimescard/en, accessed on 1 September 2022) utilized the concept of gamification to stimulate caregivers to create happiness and find meaning through card games [89].

Results also indicated there was a generation effect in terms of meaning-making [90]. Baby Boomers with a high sense of loss/powerlessness, in a combination of different levels of the sense of provision meaning, reported high depressive symptomatology (Solutions 2–4). For Gen X dementia caregivers, the presence of both a high sense of loss/powerlessness and a high sense of provision meaning contributed to high depressive symptomatology. This demarcation can be possibly explained by the fact that the heterogeneity among Baby Boomers in their life course is potentially greater than Gen X dementia caregivers. Baby Boomers may face different challenges in life, for instance, a shrinking social network after retirement, empty nest syndrome, and financial pressure after retirement. Thus, no matter the levels of a sense of provision meaning, Baby Boomers with a high sense of loss/powerlessness reported high depressive symptomatology.

On the contrary, only Gen X dementia caregivers who are more attached to their caregiving role may feel more pressure to take good care of their parents with dementia. They may find difficulties in striking a balance between different duties, for example, between work and caregiving. In this sense, those who reported a high sense of loss/powerlessness and a high sense of provision meaning were more prone to depressive symptoms. Caregivers who possessed a high sense of provision meaning may prioritize the needs of their parents with dementia over their own needs [91], which can result in depression. Future studies are required to understand the meaning-making process of dementia caregivers from two generations.

In this study, the analysis only included dementia caregivers who adapted to their caregiving roles at baseline. Prior literature showed that caregivers may adapt to the caregiving role by identifying positive aspects of providing care [92]. Our findings indicated that the adaptation period among dementia caregivers was negatively associated with depressive symptoms at follow-up. However, fsQCA results provided a closer look at this factor. It is worth noting that the adaptation period was listed in all six solutions, indicating that this is an important factor contributing to depressive symptomatology among dementia caregivers from two generations. Consistent with the bivariate analysis, Gen X dementia caregivers with a shorter adaptation period reported high depressive symptoms at follow-up. Contrarily, Baby Boomers with a longer adaptation period (Solutions 2, 4, and 5) reported high depressive symptoms at follow-up, in combination with different conditions. Future studies need to consider the adaptation period in contributing to depression among dementia caregivers, with special attention on the heterogeneity and generation among caregivers.

This study not only provides insights on adopting a generation perspective to predict depression among dementia caregivers but offers crucial hints on intervention strategies. First and foremost, dementia caregivers from both generations need to have interventions that reduce their caregiver burden and depressive symptoms. Concurrently, interventions for people with dementia are in need to reduce NPS severity. From a generation perspective, Baby Boomers dementia caregivers need support to enhance resilience through different behavioral interventions [93]. For Gen X dementia caregivers, interventions focusing on the enhancement of the sense of provisional meaning are needed. Additionally, interventions for Gen X dementia caregivers should be provided via the Internet using a self-administered approach to allow flexibility [94].

Several limitations of this study should be noted. First, the results are not generalizable to the population of dementia caregivers in other areas given the local sample and the use of a purposive sampling method. Second, the study could not establish a causal relationship between the selected conditions and depressive symptoms among dementia caregivers. However, we provided six solutions for a deeper understanding of how different combinations of conditions led to depressive symptoms among dementia caregivers from two generations. Third, this analysis only included adult child dementia caregivers who adapted to the caregiving role. Future studies should expand the scope of analysis to include caregivers who are taking care of their older family relatives and those who reported non-adaptation to caregiving roles. Fourth, this study only included dementia caregivers from two generations. With an aging population and more concern for intergeneration caregiving, future studies need to include caregivers from the Millennial generation (i.e., born between 1981 and 1996). Lastly, the fsQCA analysis strategy may not be able to establish a causal relationship between the contributing factors and depressive symptoms nor consider control variables. However, it provides a valuable way to understand different combinations of these factors and depressive symptoms which informs intervention strategies. Future studies should compare and contrast the results using traditional regression models.

## 5. Implications

This study has several implications for the understanding of the factors contributing to depressive symptomatology among dementia caregivers. First, we adopt a generation perspective to reveal the similarities and differences in developing depressive symptoms along the dementia caregiving trajectories in Baby Boomers and Gen X. This serves as a reference for future studies as it is foreseeable that dementia caregivers from other generations will join this journey in a near future, giving the increasing longevity and an aging society. Second, by adopting the fsQCA analysis and following a meaning making model, the study findings provide insights for policymakers, healthcare and social work professionals, and caregivers. Policies targeting dementia caregivers should be developed in a generation-responsive way, considering their attributions to global meaning, situational meaning, and meaning made, in line with a life course perspective. In addition, healthcare and social work professionals can draw on the findings from this study to develop and implement interventions targeting dementia caregivers from a specific generation to enhance program effectiveness. Furthermore, caregivers from different generations should be aware of potential factors that may contribute to their development of depressive symptoms and adjust their meaning making process. Lastly, this study sets an example of considering both homogeneity in the context of dementia caregiving and the heterogeneity among dementia caregivers. By this, we advocate a more inclusive approach to cater to the differential needs of dementia caregivers.

## 6. Conclusions

In summary, this study found six possible combinations of conditions leading to high depressive symptomatology among adult child dementia caregivers from two generations, the Baby Boomers and Gen X. In particular, a high caregiver burden and high depressive symptoms at baseline predicted high depressive symptomatology at follow-up, regardless of generation effect. Furthermore, the findings of this study raise a concern about the heterogeneity among Baby Boomers’ caregivers, with four different solutions predicting their development of high depressive symptomatology. Future research should examine each condition with a careful study design. Uniquely, our results indicate a need to develop interventions with a generation-responsive approach. Given the distinctions of routes leading to high depressive symptomatology among dementia caregivers from two generations, healthcare professionals and social workers should incorporate different elements into their assessments, interventions, and policy advocacy that cater to the needs of dementia caregivers. These elements should target various conditions we examined, for instance, mindfulness training to enhance resilience [85], and gamification-based intervention to enhance provisional meaning [89]. Apart from reducing the caregiver burden and depressive symptoms among dementia caregivers, it is necessary to develop more specific services from a generation perspective to enhance caregivers’ meaning-making abilities which can protect caregivers from developing depression. Interventions that enhance family resilience, respite services to relieve caregiver burden, meditation to reduce depression, as well as group interventions with caregivers from the same generation or cohort could be future possibilities. Last but not least, self-administered digital interventions to enable dementia caregivers to make meaning during their caregiving journey will be the future.

## Figures and Tables

**Figure 1 ijerph-19-15711-f001:**
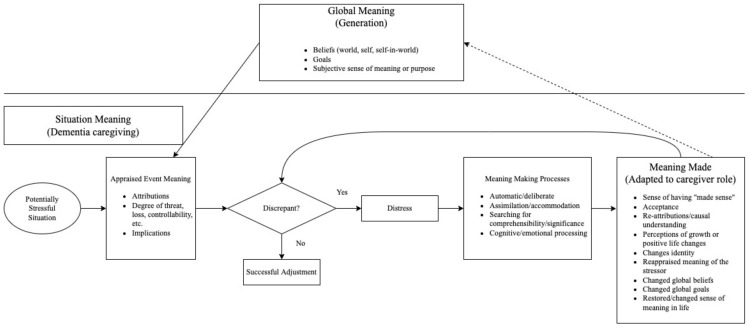
Meaning making model from a generation perspective. The model represents the meaning making processes occurred when there is a discrepancy between global and situation meaning among dementia caregivers. Meaning made is the result of the meaning making process.

**Table 1 ijerph-19-15711-t001:** Demographic characteristics of caregivers from two generations (*N* = 167).

Characteristics	Full Sample (*N* = 167)	GX (*n* = 66)	BB (*n* = 101)	Statistics	*p*
** *Caregivers* **					
Age, *M (SD)*	55.44 (6.71)	48.65 (4.10)	59.87 (3.68)	*t*(165) = −18.417	<0.001
Female, *n (%)*	132 (79.0)	53 (80.3)	79 (78.2)	*χ*^2^(1) = 0.105	0.75
Married, *n (%)*	85 (50.9)	29 (43.9)	56 (55.4)	*χ*^2^(1) = 2.115	0.15
Education, *n (%)*				*χ*^2^(1) = 0.313	0.58
Secondary of below	93 (55.7)	35 (53.0)	58 (57.4)		
Higher diploma or above	74 (44.3)	31 (47.0)	43 (42.6)		
Housing, *n (%)*				*χ*^2^(1) = 2.131	0.14
Public housing	100 (59.9)	35 (53.0)	65 (64.4)		
Private housing	67 (40.1)	31 (47.0)	36 (35.6)		
Able to afford caregiving, *n (%)*	149 (89.2)	56 (84.8)	93 (92.1)	*χ*^2^(1) = 2.170	0.14
Working, *n (%)*	89 (53.3)	45 (68.2)	44 (43.6)	*χ*^2^(1) = 9.718	0.002
** *Care recipients* **					
Age, *M (SD)*	84.76 (5.72)	81.30 (5.79)	87.02 (4.41)	*t*(165) = −0.7225	<0.001
Female, *n (%)*	131 (78.4)	47 (71.2)	84 (83.2)	*χ*^2^(1) = 3.374	0.07
Living with a caregiver, *n (%)*	92 (55.1)	38 (57.6)	54 (53.5)	*χ*^2^(1) = 0.273	0.60
No. of people living together, *M (SD)*	2.34 (1.28)	2.44 (1.19)	2.27 (1.33)	*t*(165) = 0.853	0.40
Diagnosis (years), *M (SD)*	5.02 (3.25)	4.74 (3.04)	5.21 (3.39)	*t*(165) = −0.904	0.37
ADL score, *M (SD)*	4.11 (1.99)	4.36 (1.91)	3.94 (2.03)	*t*(165) = 1.346	0.18
IADL score, *M (SD)*	1.60 (2.11)	2.00 (2.46)	1.34 (1.81)	*t*(165) = 2.006	0.05

Note. ADL = Activities of Daily Living; BB = Baby Boomers; GX = Generation X; IADL = Instrumental Activities of Daily Living; NPS = Neuropsychiatric Symptoms.

**Table 2 ijerph-19-15711-t002:** Differences in outcomes of interest among caregivers from two generations (*N* = 167).

Characteristics	Full Sample (*N* = 167)	GX (*n* = 66)	BB (*n* = 101)	Statistics	*p*
Depressive symptoms at follow-up, *M (SD)*	4.86 (5.37)	5.65 (5.95)	4.34 (4.91)	*t*(165) = 1.543	0.13
Sense of loss/powerlessness, *M (SD)*	57.31 (9.87)	59.76 (10.04)	55.70 (9.48)	*t*(165) = 2.641	0.009
Sense of provisional meaning, *M (SD)*	70.81 (9.36)	70.12 (9.48)	71.27 (9.30)	*t*(165) = −0.772	0.44
Caregiver burden, *M (SD)*	11.16 (2.90)	11.91 (2.88)	10.66 (2.83)	*t*(165) = 2.766	0.006
Depressive symptoms at baseline, *M (SD)*	3.46 (4.05)	3.48 (4.07)	3.45 (4.06)	*t*(165) = 0.061	0.95
Caregiver distress, *M (SD)*	13.71 (9.82)	14.56 (9.80)	13.16 (9.85)	*t*(165) = 0.901	0.37
Resilience, *M (SD)*	35.22 (6.58)	33.92 (6.25)	36.06 (6.68)	*t*(165) = −2.070	0.040
Adaptation period, *n (%)*				*χ*^2^(3) = 6.177	0.10
0–3 months	51 (30.5)	14 (21.2)	37 (36.6)		
4–6.5 months	36 (21.6)	18 (27.3)	18 (17.8)		
7–12 months	44 (26.3)	21 (31.8)	23 (22.8)		
13 months or above	36 (21.6)	13 (19.7)	23 (22.8)		
NPS severity, *M (SD)*	11.60 (6.77)	11.73 (6.74)	11.52 (6.82)	*t*(165) = 0.189	0.85

Note. BB = Baby Boomers; GX = Generation X; NPS = Neuropsychiatric Symptoms.

**Table 3 ijerph-19-15711-t003:** fsQCA findings towards high depressive symptomatology.

	Solution					
Configuration	1	2	3	4	5	6
Generation		●	●	●	●	⊗
** *Caregiver condition* **						
Sense of loss/powerlessness	●	●	●	●	⊗	●
Sense of provisional meaning			⊗	●	●	●
Caregiver burden	●	●	●	●	●	●
Depressive symptoms at baseline	●	●	●	●	●	●
Caregiver distress	●	⊗		●	⊗	●
Resilience	⊗	⊗	⊗			
Adaptation period	●	●	⊗	●	●	⊗
** *Care recipient condition* **						
NPS Severity	●	⊗	●	●	●	●
*Overall solution consistency*	*0.867*					
*Overall solution coverage*	*0.488*					

Note: Black circles (●) indicate the presence of a condition, and circles with “x” (⊗) indicate its absence. Blank space; “don’t care” condition. In terms of Generation, black circle (●) represents Baby Boomers, and circle with “x” (⊗) indicates Generation X.

## Data Availability

The data presented in this study are available upon reasonable request from the corresponding author.

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
