# Peer review of "Meaning Making as a Lifebuoy in Dementia Caregiving: Predicting Depression from a Generation Perspective Using a Fuzzy-Set Qualitative Comparative Analysis"

_ijerph, 2022, doi:10.3390/ijerph192315711_

Round 1

Reviewer 1 Report

Many thanks for providing me with the opportunity of reviewing the manuscript. This paper examines the predictors of depression among caregivers of people with dementia. In general, the topic of this paper is significant in the field. The structure is well written. My concerns are as follow:

Introduction

1.      Please use “patients/people with dementia” instead of “demented patients” in the manuscript. People first language should be used all over the paper.

2.      When socio-demographic and caregiver-related factors were considered, the authors may need to address more contextual issues which affect caregiving in Hong Kong or Chinese society. For example, how the impacts of culture work on adult child caregivers? Do genders matter in the field (p.2)?

3.      Like the above question, what are the meanings and social norms related to dementia caregiving in Hong Kong (p.3)?

Methods

4.      If possible, please provide evidences of psychometrics for these scales (PHQ-9, FMTCS, ZBI, NPI-Q, CD-RISC-10, Barthel Index for Activities of Daily Living, Lawton Instrumental Activities of Daily Living Scale) in Hong Kong population. Did you have local language versions of the scales?

5.      TFR or TRF (line 253-255, p.6)?

Results

6.      Could you let me know the proportion of caregivers who adapted to the caregiver role?

7.      The mean age of Baby Boomers dementia caregivers was 58.87 (line 289) or 59.87 (Table 1)?

8.      The authors should try to avoid repetition in the paragraph and the table about the descriptive findings. (p.7-8)

Discussion

9.      Regarding the differences between two generations, you may need to explain why Generation X dementia caregivers had higher sense of loss/powerlessness, higher caregiver burden (but their care recipients were younger than Baby Boomer counterparts), and lower resilience in the discussion section.

10.   When I look at Table 3, I am confused because some findings are contradictory. For example, in solution 4-6, sense of provisional meaning leads to high depressive symptoms at follow up while absence of provisional meaning leads to high depressive symptoms in solution 3. Presence of caregiver distress (solution1, 4) and absence of caregiver distress (solution 2, 5) both leads to high depressive symptoms. It is better to provide possible explanation for these phenomena. Apart from caregiver burden and depressive symptoms at baseline, other variables did not show consistent results in all solutions. Can we use the major presentation in all solutions for discussion? For example, absence (solution 2) and presence (solution 1, 3, 4, 5, 6) of high NPS severity leads to high depressive symptoms. Thus, we could say that high NPS severity contributed to high depressive symptoms for discussion? (line 394)

11.   It is reasonable to assume that high sense of loss/powerlessness leads to depression. But why sense of provision meaning could result in depression? (line 415-427)

12.   Did fsQCA in your study have any limitations you need to address in the limitations section?

13.   Although the authors adopted a generation perspective to predict depression, you may need to address different interventions for BB and GX in the manuscript.

Author Response

Reviewer 1:

1-No1#: Many thanks for providing me with the opportunity of reviewing the manuscript. This paper examines the predictors of depression among caregivers of people with dementia. In general, the topic of this paper is significant in the field. The structure is well written. My concerns are as follow:

Response:

Thank you very much for your review and feedback.

1-No2#: Introduction

Please use “patients/people with dementia” instead of “demented patients” in the manuscript. People first language should be used all over the paper.

Response:

Thank you and we amended accordingly.

“This is especially prominent among dementia caregiving as neuropsychiatric symptoms have been observed in 60% to 98% of people with dementia [7–9], and are associated with caregiver burden and depression among dementia caregivers [10–14].” (p. 2)

“Unlike their counterparts in the West, the Baby Boomer generation grew up in a Chinese cultural context and still treasures the traditional virtue of filial piety and thus willing to provide caregiving to their parents with dementia [34–36].” (p. 2)

“In this paper, the analysis only included participants who were taking care of their mother or father with dementia, self-reported adaptation to the caregiving role, and completed the questionnaire at both time points.” (p. 4)

“On the contrary, only Gen X dementia caregivers who are more attached to their caregiving role may feel more pressure to take good care of their parents with dementia.” (p. 12)

1-No3#: Introduction

When socio-demographic and caregiver-related factors were considered, the authors may need to address more contextual issues which affect caregiving in Hong Kong or Chinese society. For example, how the impacts of culture work on adult child caregivers? Do genders matter in the field (p.2)?

Response:

Thank you and we added accordingly.

“Hong Kong is expected to have more than 330,000 people aged 60 or above living with dementia [20], hence, an increase in the number of adult child dementia caregivers is a logical projection. A recent study conducted in the local context showed that women were most affected by caregiving duties, it is projected that over 60% of caregivers will be female in 2060 [21]. Another study revealed that female working dementia caregivers were more likely to feel stressed from fulfilling dual roles, which impacted mental health negatively [22]. Gender role of caregiving is especially prominent in Chinese context due to traditional value [23,24].” (p.2)

“Lastly, the fsQCA analysis strategy may not be able to establish a causal relationship between the contributing factors and depressive symptoms nor consider control variables. However, it provides a valuable way to understand different combinations of these factors and depressive symptoms which informs intervention strategies. Future studies should compare and contrast the results using traditional regression models.” (p. 13)

1-No4#: Introduction

Like the above question, what are the meanings and social norms related to dementia caregiving in Hong Kong (p.3)?

Response:

Thank you and this is similar to 1-No3#.

“Hong Kong is expected to have more than 330,000 people aged 60 or above living with dementia [20], hence, an increase in the number of adult child dementia caregivers is a logical projection. A recent study conducted in the local context showed that women were most affected by caregiving duties, it is projected that over 60% of caregivers will be female in 2060 [21]. Another study revealed that female working dementia caregivers were more likely to feel stressed from fulfilling dual roles, which impacted mental health negatively [22]. Gender role of caregiving is especially prominent in Chinese context due to traditional value [23,24].” (p.2)

1-No5#: Methods

If possible, please provide evidences of psychometrics for these scales (PHQ-9, FMTCS, ZBI, NPI-Q, CD-RISC-10, Barthel Index for Activities of Daily Living, Lawton Instrumental Activities of Daily Living Scale) in Hong Kong population. Did you have local language versions of the scales?

Response:

Thank you and we amended accordingly.

“All the questionnaires were conducted using local language, that is Traditional Chinese (in written) and Cantonese (in verbal).” (p. 5)

“The PHQ-9 has been validated in the Hong Kong population, with an internal consistency of 0.82 [54].” (p. 5)

“The ZBI-4 is extracted from the Cantonese short version of the ZBI (CZBI-Short) which has 12 items. The CZBI-Short has been validated among Hong Kong dementia caregivers, with an internal consistency of 0.84 [60].” (p. 5)

“The NPI-Q has been validated among caregivers in Hong Kong with an internal consistency of 0.76 [62].” (p. 6)

“The CD-RISC has been validated among the general population in Hong Kong, with an internal consistency of 0.97 [64].” (p. 6)

“ADL was measured by the Chinese version of the Modified Barthel Index (CMBI) to assess the care recipients’ ability to perform ten personal activities, with a potential total score ranging from 0 to 100 [49,65]. A higher score indicates a higher ability in performing ADL. The CMBI has been developed and validated in Hong Kong, with an internal consistency of 0.93 [65]. IADL was measured by the 8-item Hong Kong Chinese version of the Lawton Instrumental Activities of Daily Living Scale (CIADL), with a potential total score ranging from 0 to 8 [50,66]. A higher score indicates a higher functionality in IADL. Similar to CMBI, CIADL has been locally validated with an internal consistency of 0.86 [66].” (p. 6)

1-No6#: Methods

TFR or TRF (line 253-255, p.6)?

Response:

Thank you and we amended accordingly.

“The TFR technique relied on an empirical cumulative distribution function on the data, and it is best suited to interval-level data.” (p. 7)

1-No7#: Results

Could you let me know the proportion of caregivers who adapted to the caregiver role?

Response:

Thank you and we added accordingly.

“Of the 396 dementia caregivers enrolled in this study, a total of 167 were adult child caregivers (42.2%), with 116 (69.5%) of them adapted to caregiving role.” (p. 8)

1-No8#: Results

The mean age of Baby Boomers dementia caregivers was 58.87 (line 289) or 59.87 (Table 1)

Response:

Thank you and we amended accordingly. It is 59.87. (p. 8)

1-No9#: Results

The authors should try to avoid repetition in the paragraph and the table about the descriptive findings. (p.7-8)

Response: Thanks for the feedback. In the revised version, the repeated narratives were minimized.

“The demographic characteristics of these 167 caregivers from two generations were shown in Table 1. The mean age of Baby Boomers and Generation X dementia caregivers were significantly different (P = .001) as expected. Most of the dementia caregivers were female (79.0%), around half of them were married (50.9%), less than half of them obtained a higher diploma or above (44.3%), and around two-thirds of them were residing in public housing (59.9%), and most of them reposted that they were able to afford caregiving (89.2%). Working status among dementia caregivers from two generations was significantly different, with more Generation X dementia caregivers working at the time of baseline data collection (68.2% vs 43.6%, P = .002).” (p. 9)

1-No10#: Discussion

Regarding the differences between two generations, you may need to explain why Generation X dementia caregivers had higher sense of loss/powerlessness, higher caregiver burden (but their care recipients were younger than Baby Boomer counterparts), and lower resilience in the discussion section.

Response:

Thank you and we amended accordingly.

“Our findings indicated that Generation X dementia caregivers reported a higher sense of loss/powerlessness, a higher caregiver burden, and lower resilience than Baby Boomers dementia caregivers at baseline. Although the care recipients of Generation X dementia caregivers were significantly younger as compared to those of Baby Boomers’, we should acknowledge the fact that the deterioration rate of cognitive and functional status of people with dementia does not necessarily depends on their age. Also, nearly 70% of Generation X dementia caregivers were working. Taking care of their parents with dementia means they need to struggle between work and caregiving responsibilities, which brings negative impacts on them. For instance, they may need to sacrifice their career to provide care (e.g., switching to part-time job or forgoing promotion) but at the same time they would be worried about their financial stability [21,22,80]. Juggling between dual roles will contribute to an increase in the sense of loss/powerlessness, a higher caregiver burden and a shrink of resilience among Generation X dementia caregivers [81].” (p.12)

1-No11#: Discussion

When I look at Table 3, I am confused because some findings are contradictory. For example, in solution 4-6, sense of provisional meaning leads to high depressive symptoms at follow up while absence of provisional meaning leads to high depressive symptoms in solution 3. Presence of caregiver distress (solution1, 4) and absence of caregiver distress (solution 2, 5) both leads to high depressive symptoms. It is better to provide possible explanation for these phenomena. Apart from caregiver burden and depressive symptoms at baseline, other variables did not show consistent results in all solutions. Can we use the major presentation in all solutions for discussion? For example, absence (solution 2) and presence (solution 1, 3, 4, 5, 6) of high NPS severity leads to high depressive symptoms. Thus, we could say that high NPS severity contributed to high depressive symptoms for discussion? (line 394)

Response:

Thank you and we amended accordingly.

“Among five solutions (except Solution 2), the presence of high NPS severity in combination with other conditions contributed to high depressive symptomatology among dementia caregivers. This result is consistent with prior literature that showed a relationship between caregiver burden, depressive symptoms, and dementia patients’ NPS severity [83]. The results confirmed the positive association between NPS severity and caregiver distress but also revealed distinctive results from a generation perspective. In particular, high NPS severity may lead to high depressive symptomatology among Baby Boomers' dementia caregivers in the absence of caregiver distress.” (p. 12)

1-No12#: Discussion

It is reasonable to assume that high sense of loss/powerlessness leads to depression. But why sense of provision meaning could result in depression? (line 415-427)

Response:

Thank you and we added accordingly.

“On the contrary, only Gen X dementia caregivers who are more attached to their caregiving role may feel more pressure to take good care of their parents with dementia. They may find difficulties in striking a balance between different duties, for example, between work and caregiving. In this sense, those who reported a high sense of loss/powerlessness and a high sense of provision meaning were more prone to depressive symptoms. For caregivers who possessed a high sense of provision meaning may pri-oritize the needs of their parents with dementia than their own needs [91], which can result in depression depression. Future studies are required to understand the meaning-making process of dementia caregivers from two generations.” (p. 13)

1-No13#: Discussion

Did fsQCA in your study have any limitations you need to address in the limitations section?

Response:

Thank you and we added accordingly.

“Lastly, the fsQCA analysis strategy may not be able to establish a causal relationship between the contributing factors and depressive symptoms nor consider control variables. However, it provides a valuable way to understand different combinations of these factors and depressive symptoms which informs intervention strategies. Future studies should compare and contrast the results using traditional regression models.” (p. 13)

1-No14#: Discussion

Although the authors adopted a generation perspective to predict depression, you may need to address different interventions for BB and GX in the manuscript.

Response:

Thank you and we added accordingly.

“This study not only provides insights on adopting a generation perspective to predict depression among dementia caregivers but offers crucial hints on intervention strategies. First and the foremost, dementia caregivers from both generations need to have interventions that reduce their caregiver burden and depressive symptoms. Concurrently, interventions for people with dementia are in need to reduce NPS severity. From a generation perspective, Baby Boomers dementia caregivers need support to enhance resilience through different behavioral interventions [93]. For Generation X dementia caregivers, interventions focusing on enhancement of the sense of provisional meaning is needed. Also, interventions for Generation X dementia caregivers should be provided via the Internet using a self-administered approach to allow flexibility [94].” (p. 13)

Reviewer 2 Report

Meaning making as a lifebuoy in dementia caregiving: Predicting depression from a generation perspective using a fuzzy-set qualitative comparative analysis

REVIEW

Thanks for the opportunity to review this important paper. The topic is timely and highly relevant for caregiving health. Hereinafter are some suggestions to improve the manuscript.

INTRODUCTION

o    This section is nicely written. However, I suggest giving a definition of a few key concepts that the reader will have to make sense of. These key concepts that need to be elaborated and accompanied by appropriate references are:

        §   Concept of meaning making, that is the ability to transform meaning;

        §   Concept of situational meaning, that is the specific evaluation of an event;

        §   Concept of meaning made, that is the positive outcome enhancing wellbeing and coping.

        §   These three concepts should then be described together (e.g., when there is a discrepancy between global and situational meaning, a meaning making process occurs..)

o    The gap in the literature should be more explicitly and clearly outlined.

o    Figure 1 should be accompanied by a brief explanation of the model. So, for example: Figure 1. Meaning making model from a generation perspective. The model represents …

o    The concept of generation perspective should be more elaborated. Furthermore, a rationale for choosing the two generations should be offered.

o    Fuzzy-set qualitative comparative analysis should be briefly defined (above and beyond references).

MATERIALS AND METHODS

o    The authors wrote: “In this paper, the analysis only included participants who were taking care of their demented mother or father, self-reported adaptation to the caregiving role, and completed the questionnaire at both time points”. This makes me think that the caregivers are not affected by dementia, but the patients are. Please clarify.

o    Please give info on the psychometric properties of the instruments used.

o    This sentence: “generation affects global meaning” seems to be out of place. This aspect should be better explained in the intro.

o    In the paragraph “situational meaning”, please give a definition of provisional meaning (line 166).

o    In line 217, please add “respectively” at the end.

o    Consider moving some content of the paragraph 2.4.2 to the introduction, especially the novelty of the analysis and the gap of the literature. If possible, consider shortening the paragraph further, as it is very hard to follow.

RESULTS

o    Very clear

DISCUSSION

o    Well done

CONCLUSIONS

o    Please be synthetic and do not repeat the results

I wish all authors well on the revisions

Author Response

Reviewer 2:

2-No1#: Thanks for the opportunity to review this important paper. The topic is timely and highly relevant for caregiving health. Hereinafter are some suggestions to improve the manuscript.

Response:

Thank you very much for your review and feedback.

2-No2#: Introduction

This section is nicely written. However, I suggest giving a definition of a few key concepts that the reader will have to make sense of. These key concepts that need to be elaborated and accompanied by appropriate references are:

  • Concept of meaning making, that is the ability to transform meaning;

Response:

Thank you and we amended accordingly.

“Meaning is defined as the “mental representation of possible relationships among things, events, and relationships” [31]. A phenomenal integrative review of meaning making provided a systematic way to understand how individuals make sense of stressful life events [32]. This is the ability for an individual to transform meaning.” (p.2)

2-No3#: Introduction

  • Concept of situational meaning, that is the specific evaluation of an event;

Response:

Thank you and we amended accordingly.

“Potentially stressful events, such as dementia caregiving, will challenge the global meaning of individuals. Under this circumstance, individuals will appraise difficult situations and assign meanings to them [33]. This means a specific evaluation of an event.” (p. 2)

2-No4#: Introduction

  • Concept of meaning made, that is the positive outcome enhancing wellbeing and coping.

Response:

Thank you and we amended accordingly.

“To minimize the discrepancies between situational meaning and global meaning, indi-viduals engage in meaning-making processes, with the results or changes being termed “meaning made” [32]. This is the positive outcome to cope with the stressful events and enhance wellbeing.” (p. 2)

2-No5#: Introduction

  • These three concepts should then be described together (e.g., when there is a discrepancy between global and situational meaning, a meaning making process occurs..)

Response:

Thank you and we amended accordingly.

“This is the positive outcome to cope with the stressful events and enhance wellbeing. In this sense, when there is a discrepancy between global and situation meaning, the meaning-making processes occur to provide a new appraisal of the stressful events [32].” (p. 2

2-No6#: Introduction

The gap in the literature should be more explicitly and clearly outlined.

Response:

Thank you and we amended accordingly.

“There are three major gaps in extant literature. First, there is a lack of generational approach to understand dementia caregivers, despite it has been investigated and incorporated into different disciplines, such as marketing [43], technology [44], and healthcare [45]. Second, positive aspects in caregiving such as searching for meaning is insufficient. Recently, several scholars sought to examine differences in caregiver mental health outcomes across generations [46], however, they only focus only on the negative aspects of dementia caregiving. Third, heterogeneity among dementia caregivers was disregarded using conventional analysis strategies. What is lacking and urgently needed is empirical evidence to unveil conditions contributing to depressive symptoms among dementia caregivers. Guided by the meaning making model, this study examined the relationships between meaning making, generation, and high depressive symptomatology among dementia caregivers, using a configurational approach called fuzzy-set qualitative comparative analysis [47,48].” (p. 3-4)

2-No7#: Introduction

Figure 1 should be accompanied by a brief explanation of the model. So, for example: Figure 1. Meaning making model from a generation perspective. The model represents …

Response:

Thank you and we amended accordingly.

“Figure 1. Meaning making model from a generation perspective. The model represents the meaning making processes occurred when there is a discrepancy between global and situation meaning among dementia caregivers. Meaning made is the result of the meaning making process. (p. 3)

2-No8#: Introduction

The concept of generation perspective should be more elaborated. Furthermore, a rationale for choosing the two generations should be offered.

Response:

Thank you and we amended accordingly.

“Baby Boomers were born following World War II and are considered to be the first generation to enjoy a comparatively higher educational attainment and wealth. Family structures and experiences among the Baby Boomer generation evolved in an unprec-edented manner, including an increased divorce rate and a soaring number of women entering the workforce [38]. Unlike their counterparts in the West, the Baby Boomers generation grew up in a Chinese cultural context and still treasures the traditional virtue of filial piety and thus willing to provide caregiving to their parents with dementia [39–41]. Generation X (Gen X), composed of the next generation of Baby Boomers, was sig-nified by flexibility, independence, and self-reliant traits. As a generation, Gen X has a different global view as compared to the Baby Boomer generation and places high value on work-life balance which allows them to focus on family as well as personal well-being [42]. Due to the soaring number of dementia caregivers in Hong Kong, it is expected these two generations will be particularly prone to depression due to caregiving duties as well as experiencing aging themselves. In this sense, we purposefully selected Baby Boomers and Gen X in our study [21,22].” (p. 3)

2-No9#: Introduction

Fuzzy-set qualitative comparative analysis should be briefly defined (above and beyond references).

Response:

Thank you and we amended accordingly.

“In this study, we are applying a generation perspective to examine the factors leading to depressive symptomatology among Baby Boomers and Gen X dementia caregivers, traditional analysis strategies are eloquent of an assumption of homogeneity of individuals from the same generation. To unveil different possibilities that contributed to depressive symptoms during the dementia caregiving journey, we should embrace the heterogeneity of dementia caregivers who had different individual experiences from a life course perspective by employing the fuzzy-set qualitative comparative analysis (fsQCA). fsQCA is originated in the social sciences field to combine case-oriented and varia-ble-oriented quantitative analysis, it started with the creation of qualitative comparative analysis, drawing on the fuzzy-set theory [68]. Guided by the meaning making model, this study examined the relationships between meaning making, generation, and high depressive symptomatology among dementia caregivers, using the fsQCA [47,48].” (p. 4)

2-No10#: Materials and methods

The authors wrote: “In this paper, the analysis only included participants who were taking care of their demented mother or father, self-reported adaptation to the caregiving role, and completed the questionnaire at both time points”. This makes me think that the caregivers are not affected by dementia, but the patients are. Please clarify.

Response:

Thank you and we added accordingly.

“These participants are not affected by dementia themselves but taking care of parents with dementia.” (p. 4)

2-No11#: Materials and methods

Please give info on the psychometric properties of the instruments used.

Response:

This is similar to 1-No5#.

“The PHQ-9 has been validated in the Hong Kong population, with an internal consistency of 0.82 [54].” (p. 5)

“The ZBI-4 is extracted from the Cantonese short version of the ZBI (CZBI-Short) which has 12 items. The CZBI-Short has been validated among Hong Kong dementia caregivers, with an internal consistency of 0.84 [60].” (p. 5)

“The NPI-Q has been validated among caregivers in Hong Kong with an internal consistency of 0.76 [62].” (p. 6)

“The CD-RISC has been validated among the general population in Hong Kong, with an internal consistency of 0.97 [64].” (p. 6)

“ADL was measured by the Chinese version of the Modified Barthel Index (CMBI) to assess the care recipients’ ability to perform ten personal activities, with a potential total score ranging from 0 to 100 [49,65]. A higher score indicates a higher ability in performing ADL. The CMBI has been developed and validated in Hong Kong, with an internal consistency of 0.93 [65]. IADL was measured by the 8-item Hong Kong Chinese version of the Lawton Instrumental Activities of Daily Living Scale (CIADL), with a potential total score ranging from 0 to 8 [50,66]. A higher score indicates a higher functionality in IADL. Similar to CMBI, CIADL has been locally validated with an internal consistency of 0.86 [66].” (p. 6)

2-No12#: Materials and methods

This sentence: “generation affects global meaning” seems to be out of place. This aspect should be better explained in the intro.

Response:
Thank you and we amended accordingly.

“Generation is the proxy of global meaning.” (p. 4)

2-No13#: Materials and methods

In the paragraph “situational meaning”, please give a definition of provisional meaning (line 166).

Response:

Thank you and we amended accordingly.

“Situational meaning is the specific evaluation of a potential stressful event. It was measured by two subscales, loss/powerlessness and provisional meaning, of the Finding Meaning Through Caregiving Scale (FMTCS) [57].” (p. 4)

2-No14#: Materials and methods

In line 217, please add “respectively” at the end.

Response:

Thank you and we added accordingly.

“Ethics approval was obtained from the Human Research Ethics Committee, The University of Hong Kong as well as the Institutional Review Board of The University of Hong Kong/Hospital Authority Hong Kong West Cluster, respectively.” (p. 4)

2-No15#: Materials and methods

Consider moving some content of the paragraph 2.4.2 to the introduction, especially the novelty of the analysis and the gap of the literature. If possible, consider shortening the paragraph further, as it is very hard to follow.

Response:

Thank you and we amended accordingly.

“In this study, we are applying a generation perspective to examine the factors leading to depressive symptomatology among Baby Boomers and Gen X dementia caregivers, traditional analysis strategies are eloquent of an assumption of homogeneity of individuals from the same generation. To unveil different possibilities that contributed to depressive symptoms during the dementia caregiving journey, we should embrace the heterogeneity of dementia caregivers who had different individual experiences from a life course perspective by employing the fuzzy-set qualitative comparative analysis (fsQCA). fsQCA is originated in the social sciences field to combine case-oriented and varia-ble-oriented quantitative analysis, it started with the creation of qualitative comparative analysis, drawing on the fuzzy-set theory [68]. Guided by the meaning making model, this study examined the relationships between meaning making, generation, and high depressive symptomatology among dementia caregivers, using the fsQCA [47,48].” (p. 4)

“2.4.2. Fuzzy-set Qualitative Comparative Analysis of Depressive Symptoms

While the development of fsQCA is still in its infancy, the number of researchers across disciplines who use fsQCA has steadily increased [69]. This paper is the first to apply fsQCA to understand the factors contributing to depressive symptomatology among dementia caregivers. One of the main characteristics of fsQCA is the degrees of freedom it provided to allow researchers to examine the causal complexities involved in conjunctural causation [70]. Fundamentally, the aim of using fsQCA is to examine the cases configurationally, meaning that separate parts of a whole picture have to be understood in a case-oriented manner, embedded in specific context such as in a dementia caregiving journey. In addition, by following the meaning making model, we look into various aspects and combinations of conditions to obtain a comprehensive understanding of the homogeneity within the context. In general, by applying fsQCA and following the meaning making model, this analysis strategy allows us to depict the combinations (also called solutions in fsQCA) to examine the factors leading to depressive symptomatology among dementia caregivers from two generations, while not losing sight in both heterogeneity and homogeneity.

Configurational analysis was performed using SPSS version 27 (IBM®), QCA package in R version 4.2.0[71], and fsQCA software version 3.0 at different steps. Key analytic fsQCA steps include analyzing contrarian cases, choosing thresholds, calibrating data, constructing a truth table, sorting the truth table, computing solutions, and identifying predictive validity [72,73].

First, we conducted a contrarian case analysis in SPSS to identify the main effects and contrarian cases (Supplement 1). Next, we performed data calibration in R to transform the antecedent conditions and outcomes into a membership score ranging from 0 to 1. Specifically, membership scores of 0 and 1 indicate non-full and full membership, respectively. A score of 0.5 denotes intermediate set membership. In this study, we used the totally fuzzy and relative (TFR) method [74]. Data calibration using the TRF method was based on rank orders, it is particularly applicable for this study as we categorize dementia caregivers into two generations and comprised both ordinal and interval levels data. The TFR technique relied on an empirical cumulative distribution function on the data, and it is best suited to interval-level data. A normalized version by applying a simple transformation to create a membership score was applied to categorical data. We followed prior literature in applying the below formula for data calibration [75]. In which, E() is the empirical cumulative distribution function of the data.

After data calibration, we constructed a truth table with 2k rows using the fsQCA software, where k denotes the number of antecedent conditions. In this study, the truth table involves 29 logically possible combinations of antecedent conditions. The truth table was then minimized by the number of cases smaller than 1, a raw consistency value below 0.75, and a proportional reduction in inconsistency below 0.70 [72], following the Quine-McCluskey algorithm (i.e., minimizing Boolean functions) [76–78].

The fsQCA provides three types of solutions, namely complex, parsimonious, and intermediate [47]. The complex solution presents all possible combinations when a logical operation (i.e., choosing the presence or absence/negation of a variable) is applied. The parsimonious solution is a simplified version of the complex solution, it only presents the “core conditions” (i.e., conditions indicate a strong causal relationship with the outcome). The intermediate solution is retrieved following a counterfactual analysis of the complex and the parsimonious solutions, it only presents theoretically plausible counterfactuals [47]. It is important to note that while “core conditions” are presented in both parsimonious and intermediate solutions, conditions that are eliminated in the parsimonious solution but appear only in the intermediate solution are called “peripheral conditions” (i.e., conditions indicate a weaker causal relationship with the outcome) [47]. In this sense, we identified the “core conditions” by examining the parsimonious solution and included the “peripheral conditions” by referring to the intermediate solution. This strategy of combining the parsimonious and intermediate solutions offers an aggregated view of the findings and provides richer information for interpretation [47]. (p. 6-8)

2-No16#: Results

Very clear

Response:

Thank you very much for your review and feedback.

2-No17#: Discussion

Well done

Response:

Thank you very much for your review and feedback.

2-No18#: Conclusions

Please be synthetic and do not repeat the results

Response:

Thank you and we amended accordingly.

“In summary, this study found six possible combinations of conditions leading to high depressive symptomatology among adult child dementia caregivers from two genera-tions, the Baby Boomers and Gen X. In particular, a high caregiver burden and high depressive symptoms at baseline predicted high depressive symptomatology at fol-low-up, regardless of generation effect. Furthermore, the findings of this study raise the concern about the heterogeneity among Baby Boomers' caregivers, with four different solutions predicting their development of high depressive symptomatology. Future re-search should examine each condition with a careful study design. Uniquely, our results indicate a need to develop interventions with a generation-responsive approach. Given the distinctions of routes leading to high depressive symptomatology among dementia caregivers from two generations, healthcare professionals and social workers should incorporate different elements into their assessments, interventions, and policy advocacy that cater to the needs of dementia caregivers. These elements should target various conditions we examined, for instance, mindfulness training to enhance resilience [85], gamification-based intervention to enhance provisional meaning [89]. Apart from re-ducing the caregiver burden and depressive symptoms among dementia caregivers, it is necessary to develop more specific services from a generation perspective to enhance caregivers’ meaning-making abilities which can protect caregivers from developing depression. Interventions that enhance family resilience, respite services to relieve caregiver burden, meditation to reduce depression, as well as group interventions with caregivers from the same generation or cohort could be future possibilities. Last but not least, self-administered digital interventions to enable dementia caregivers to make meaning during their caregiving journey will be the future.” (p. 14)

2-No19#: I wish all authors well on the revisions

Response:

Thank you very much.

Round 2

Reviewer 2 Report

Thanks for following my recommendations. All the issues have adequately been addressed. Well done!